# Reinforcement Learning with Latent Flow

**Wenling Shang**[*]
DeepMind
wendyshang@deepmind.com

**Xiaofei Wang**[*]
UC Berkeley
w.xf@berkeley.edu

**Aravind Srinivas**
OpenAI
aravind_srinivas@berkeley.edu

**Aravind Rajeswaran**
Facebook AI Research, University of Washington
aravraj@fb.com

**Yang Gao**
Tsinghua University
gaoyangiiis@tsinghua.edu.cn

**Pieter Abbeel**
UC Berkeley, Covariant
pabbeel@berkeley.edu

**Michael Laskin**
UC Berkeley
mlaskin@berkeley.edu

## Abstract

Temporal information is essential to learning effective policies with Reinforcement Learning (RL). However, current state-of-the-art RL algorithms either assume that such information is given as part of the state space or, when learning from pixels, use the simple heuristic of frame-stacking to implicitly capture temporal information present in the image observations. This heuristic is in contrast to the current paradigm in video classification architectures, which utilize explicit encodings of temporal information through methods such as optical flow and two-stream architectures to achieve state-of-the-art performance. Inspired by leading video classification architectures, we introduce the **F**low of **La**tents for **Re**inforcement Learning (*Flare*), a network architecture for RL that explicitly encodes temporal information through latent vector differences. We show that Flare recovers optimal performance in state-based RL without explicit access to the state velocity, solely with positional state information. Flare is the most sample efficient model-free pixel-based RL algorithm on the DeepMind Control suite when evaluated on the 500k and 1M step benchmarks across 5 challenging control tasks, and, when used with Rainbow DQN, outperforms the competitive baseline on Atari games at 100M time step benchmark across 8 challenging games.

## 1 Introduction

Reinforcement learning (RL) [41] holds the promise of enabling artificial agents to solve a diverse set of tasks in uncertain and unstructured environments. Recent developments in RL with deep neural networks have led to tremendous advances in autonomous decision making. Notable examples include classical board games [36, 37], video games [29, 6, 45], and continuous control [34, 28]. There has been a large body of research on extracting high quality features during the RL process, such as with auxiliary losses [20, 27, 35] or data augmentation [25, 26]. However, another important component in RL representation learning has been largely overlooked: a more effective architecture to incorporate temporal features. This becomes especially crucial in an unstructured real-world setup like the home when compact state representations such as calibrated sensory inputs are unavailable. Motivated by this understanding, we explore architectural improvements to better utilize temporal features for the problem of efficient and effective deep RL from pixels.

---

[*]Equal contribution

35th Conference on Neural Information Processing Systems (NeurIPS 2021).

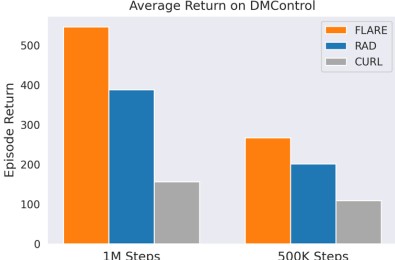 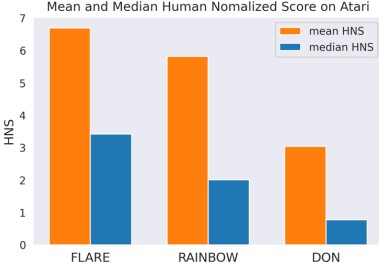

Figure 1: Mean and median evaluation scores on DMControl[42] and Atari[3]. Flare is an architectural modification that improves RAD and Rainbow, the base algorithms it integrates with.

Current approaches in deep RL for learning temporal features are largely heuristic in nature. A commonly employed approach is to stack the most recent frames [29] as inputs to a convolutional neural network (CNN). This can be interpreted as a form of early fusion [24], where information from the recent time window is combined at the pixel level for input to the CNN. In contrast, modern video recognition systems use alternate architectures that employ optical flow and late fusion [38], where frames are processed individually with CNN layers before fusion and downstream processing. Late fusion is typically beneficial due to better performance, fewer parameters, and the ability to use multi-modal data [8, 21]. However, it is not straightforward how to directly extend such architectures to RL. Real-time computation of optical flow for action selection can be computationally infeasible for many applications with fast control loops like robotics. Furthermore, optical flow computation at training time can also be prohibitively expensive. In our experiments, we also observe that a naive late fusion architecture minus the optical flow yields poor results in RL settings (see Section 6.3). This observation is consistent with recent findings in related domains like visual navigation [46].

To overcome the above challenges, we develop **F**low of **La**tents for **Re**inforcement Learning (*Flare*), a new architecture for deep RL from pixels (Figure 4). Flare can be interpreted as a *structured late fusion* architecture. It processes each frame individually to compute latent vectors, similar to a standard late fusion approach. Subsequently, temporal differences between the latent feature vectors are computed and fused along with the latent vectors by concatenation for downstream processing. By incorporating this structure of temporal difference in latent feature space, we provide the learning agent with appropriate inductive bias.

We highlight the main empirical contributions from Flare in the following[2]:

1. Flare recovers optimal performance in state-based RL without explicit access to the state velocity, solely with positional state information.

2. Flare achieves state-of-the-art performance compared to model-free methods on several *challenging* pixel-based continuous control tasks within the DeepMind control benchmark suite [42], while being the most sample efficient model-free pixel-based RL algorithm across these tasks, outperforming the prior model-free state-of-the-art RAD on the 500k and 1M environment step benchmarks respectively (Figure 1). A video demonstration of Flare achieving SOTA performance on Quadruped Walk is in the supplementary materials.

3. When augmented over Rainbow DQN, Flare outperforms the baseline on 5 out of 8 challenging Atari games at 100M step benchmark. Notably, Flare scores 1668 on Montezuma's Revenge, a signifcant gain over the baseline Rainbow DQN's 900.

## 2 Related Work

**Pixel-Based RL** The ability of an agent to autonomously learn control policies from visual inputs can greatly expand the applicability of deep RL [10, 32]. Prior works have used CNNs to extend RL algorithms like PPO [34], SAC [14], and Rainbow [19] to pixel-based tasks. Such direct extensions have typically required substantially larger number of environment interactions when compared to the state-based environments. In order to improve sample efficiency, recent efforts have studied the use of auxiliary tasks and loss functions [50, 27, 35], data augmentation [26, 25], and latent space

---

[2]Code: https://github.com/WendyShang/flare

dynamics modeling [16, 15]. Despite these advances, there is still a large gap between the learning efficiency in state-based and pixel-based environments in a number of challenging benchmark tasks. Our goal in this work is to identify where and how to improve pixel-based performance on this set of challenging control environments.

**Neural Network Architectures in RL** Mnih et al. [29] combined Q-learning with CNNs to achieve human level performance in Atari games, wherein Mnih et al. [29] concatenate the most recent 4 frames and use a convolutional neural network to output the Q values. In 2016, Mnih et al. [30] proposed to use a shared CNN among frames to extract visual features and aggregate the temporal information with LSTM. The same architectures have been adopted by most works to date [27, 35, 25, 26]. Recently, new architectures for RL have been explored that explore dense connections [39] as well as residual connections and instance norm [49]. However, the development of new architectures to better capture temporal information in a stream of images has received little attention in deep RL, and our work fills this void. Perhaps the closest to our motivation is the work of Amiranashvili et al. [1] who explicitly use optical flow as an extra input to the RL policy. However, this approach requires additional information and supervision signal to train the flow estimator, which could be unavailable or inaccurate in practice. In contrast, our approach is a simple modification to existing deep RL architectures and does not require any additional auxiliary tasks or supervision signals.

**Two-Stream Video Classification** In video classification tasks, such as activity recognition [40], there are a large body of works on how to utilize temporal information [9, 22, 44, 7, 47, 12]. Of particular relevance is the two-stream architecture of Simonyan and Zisserman [38], where one CNN stream takes the usual RGB frames, while the other the optical flow computed from the RGB values. The features from both streams are then late-fused to predict the activity class. That the two-stream architecture yields a significant performance gain compared to the single RGB stream counterpart, indicating the explicit temporal information carried by the flow plays an essential role in video understanding. Instead of directly computing the optical flow, we propose to capture the motion information in latent space to avoid computational overheads and potential flow approximation errors. Our approach also could focus on domain-specific motions that might be overlooked in a generic optical flow representation.

## 3 Background

**Soft Actor Critic** (SAC) [14] is an off-policy actor-critic RL algorithm for continuous control with an entropy maximization term augmented to its score function to encourage exploration. SAC learns a policy network $\pi_\psi(a_t|\mathbf{o}_t)$ and critic networks $Q_{\phi_1}(\mathbf{o}_t, a_t)$ and $Q_{\phi_2}(\mathbf{o}_t, a_t)$ to estimate state-action values. The critic $Q_{\phi_i}(\mathbf{o}_t, a_t)$ is optimized to minimize the (soft) Bellman residual error:

$$\mathcal{L}_Q(\phi_i) = \mathbb{E}_{\tau \sim \mathcal{B}} \left[ \left( Q_{\phi_i}(\mathbf{o}_t, a_t) - (r_t + \gamma V(\mathbf{o}_{t+1})) \right)^2 \right],$$

where $r$ is the reward, $\gamma$ the discount factor, $\tau = (\mathbf{o}_t, a_t, \mathbf{o}_{t+1}, r_t)$ is a transition sampled from replay buffer $\mathcal{B}$, and $V(\mathbf{o}_{t+1})$ is the (soft) target value estimated by:

$$V(\mathbf{o}_{t+1}) = \min_i Q_{\bar{\phi}_i}(\mathbf{o}_{t+1}, a_{t+1}) - \alpha \log \pi_\psi(a_{t+1}|\mathbf{o}_{t+1})],$$

where $\alpha$ is the entropy maximization coefficient. For stability, $Q_{\bar{\phi}_i}$ is the exponential moving average of $Q_{\phi_i}$'s over training iterations. The policy $\pi_\psi$ is trained to maximize the expected return estimated by $Q$ together with the entropy term

$$L_\pi(\psi) = -\mathbb{E}_{a_t \sim \pi} \left[ \min_i Q_{\phi_i}(\mathbf{o}_t, a_t) - \alpha \log \pi_\psi(a_t|\mathbf{o}_t) \right],$$

where $\alpha$ is also a learnable parameter.

**Reinforcement Learning with Augmented Data**, or RAD [26], is a recently proposed training technique. In short, RAD pre-processes raw pixel observations by applying random data augmentations, such as random translation or cropping, for RL training. As simple as it is, RAD has taken many existing RL algorithms, including SAC, to the next level. For example, on many DMControl [42] benchmarks, while vanilla pixel-based SAC performs poorly, RAD-SAC—i.e. applying data augmentation to pixel-based SAC—achieves state-of-the-art results both in sample efficiency and final performance. In this work, we refer RAD to RAD-SAC and use random translation as data augmentation.

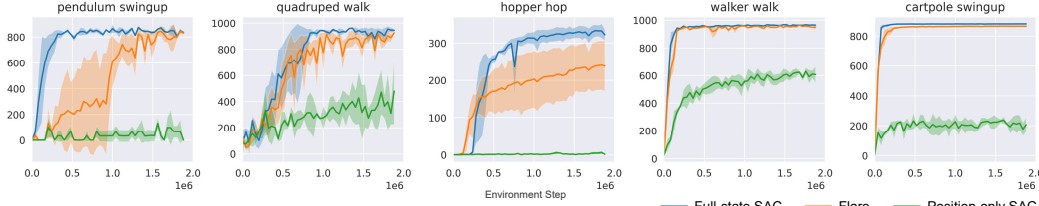

Figure 2: Flare enables an RL agent with only access to positional state to recover a near-optimal policy relative to RL with access to the full state. In the above learning curves we show test-time performance for (i) full-state SAC (blue), where both pose and temporal information is given (ii) position-only SAC (green), and (iii) state-based Flare (orange), where only pose information is provided and velocities are approximated through pose offsets. Unlike full-state SAC, which learns the optimal policy, position-only SAC either fails or converges at suboptimal policies. Meanwhile, the fusion of positions and approximated velocities in Flare efficiently recovers near-optimal policies in most cases. This motivates using Flare for pixel-based input, where velocities are not present in the observation. These results show mean performance with standard deviations averaged over 3 seeds.

**Rainbow DQN** is an extension of the Deep Q Network (DQN) [29], which combines multiple follow-up improvements of DQN to a single algorithm [19]. In summary, DQN [29] is an off-policy RL algorithm that leverages deep neural networks (DNN) to estimate the Q value directly from the pixel space. The follow-up works Rainbow DQN bring together to enhance the original DQN include double Q learning [17], prioritized experience replay [33], dueling network [48], noisy network [13], distributional RL [5] and multi-step returns [41]. Rainbow DQN is one of the state-of-the-art RL algorithms on the Atari 2600 benchmark [3]. We thus adopt an official implementation of Rainbow [31] as our baseline to directly augment Flare on top.

## 4 Motivation

We motivate Flare by investigating the importance of temporal information in state-based RL. Our investigation utilizes 5 diverse DMControl [42] tasks. The full state for these environments includes both the agent's pose information, such as the joints' positions and angles, as well as temporal information, such as the joints' translational and angular velocities.

First, we train two variants with SAC—one where the agent receives the full state as input (full-state SAC), and the other with the temporal information masked out, i.e. the agent only receives the pose information as its input (position-only SAC). The resulting learning curves are in Figure 2. While the full-state SAC learns the optimal policy quickly, the position-only SAC learns much sub-optimal policies, which often fail entirely. Therefore, we conclude that effective policies cannot be learned from positions alone, and that temporal information is crucial for efficient learning.

While full-state SAC can receive velocity information from internal sensors in simulation, in the more general case such as learning from pixels, such information is often not readily available. For this reason, we attempt to approximate temporal information as the difference between two consecutive states' positions. Concretely, we compute the positional offset $\delta_t = (s_t^p - s_{t-1}^p, s_{t-1}^p - s_{t-2}^p, s_{t-2}^p - s_{t-3}^p)$, and provide the fused vector $(s_t^p, \delta_t)$ to the SAC agent. This procedure describes the state-based version of Flare. Results shown in Figure 2 demonstrate that state-based Flare significantly outperforms the position-only SAC. Furthermore, it achieves optimal asymptotic performance and a learning efficiency comparable to full-state SAC in most environments.

Given that the position-only SAC utilizes $s_t^p$ compared to Flare that utilizes $s_t^p$ and $\delta_t$, we also investigate a variant (stack SAC) where the SAC agent takes consecutive positions $(s_t^p, s_{t-1}^p, s_{t-2}^p, s_{t-3}^p)$. Stack SAC reflects the frame-stack heuristic used in pixel-based RL. Results in Figure 3 show that Flare still significantly outperforms stack SAC. It suggests that the well-structured inductive bias in the form of temporal-position fusion is essential for efficient learning.

Lastly, since a recurrent structure is an alternative approach to process temporal information, we implement an SAC variant with recurrent modules (Recurrent SAC) to compare with Flare. Specifically,

we pass a sequence of poses $s_t^p, s_{t-1}^p, s_{t-2}^p, s_{t-3}^p$ through an LSTM cell. The number of the LSTM hidden units $h$ is set to be the same as the dimension of $\delta_t$ in Flare. The trainable parameters of the LSTM cell are updated to minimize the critic loss. Recurrent SAC is more complex to implement and requires longer wall-clock training time, but performs worse than Flare as shown in Figure 3.

Our findings from the state experiments in Figure 2 and Figure 3 suggest that (i) temporal information is crucial to learning effective policies in RL, (ii) using Flare to approximate temporal information in the absence of sensors that provide explicit measurements is sufficient in most cases, and (iii) to incorporate temporal information via naively staking position states or a recurrent module are less effective than Flare. In the next section, we carry over these insights to pixel-space RL.

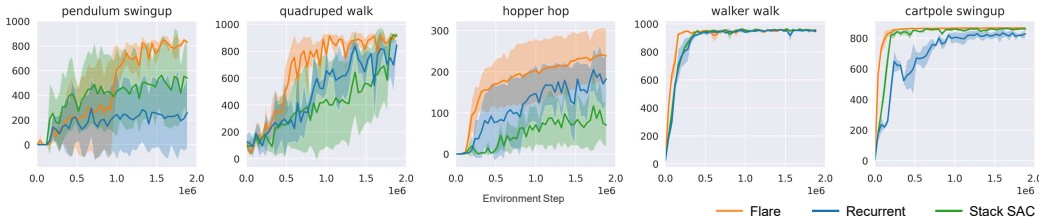

Figure 3: We compare 3 ways to incorporate temporal information: i) Flare (orange) receives $(s_t^p, s_t^p - s_{t-1}^p, s_{t-1}^p - s_{t-2}^p, s_{t-2}^p - s_{t-3}^p)$, ii) stack SAC (green) stacks $(s_t^p, s_{t-1}^p, s_{t-2}^p, s_{t-3}^p)$ as inputs, and iii) recurrent SAC (blue) uses recurrent layers to process $(s_t^p, s_{t-1}^p, s_{t-2}^p, s_{t-3}^p)$. Stack SAC and recurrent SAC perform significantly worse than Flare on most environments, highlighting the benefit of how Flare handles temporal information. Results are averaged over 3 seeds.

## 5 Reinforcement Learning with Latent Flow

To date, frame stacking is the most common way of pre-processing pixel-based input to convey temporal information for RL algorithms. This heuristic, introduced by Mnih et al. [29], has been largely untouched since its inception and is used in most state-of-the-art RL architectures. However, our observations from the experiments run on state inputs in Section 4 suggest an alternative to the frame stacking heuristic through the explicit inclusion of temporal information as part of the input. Following this insight, we seek a general alternative approach to explicitly incorporate temporal information that can be coupled to any base RL algorithm with minimal modification. To this end, we propose the **F**low of **La**tents for **Re**inforcement Learning (*Flare*) architecture. Our proposed method calculates differences between the latent encodings of individual frames and fuses the feature differences and latent embeddings before passing them as input to the base RL algorithm, as shown in Figure 4. We demonstrate Flare on top of 2 state-of-the-art model-free off-policy RL baselines, RAD-SAC [26] and Rainbow DQN [19], though in principle any RL algorithm can be used in principle.

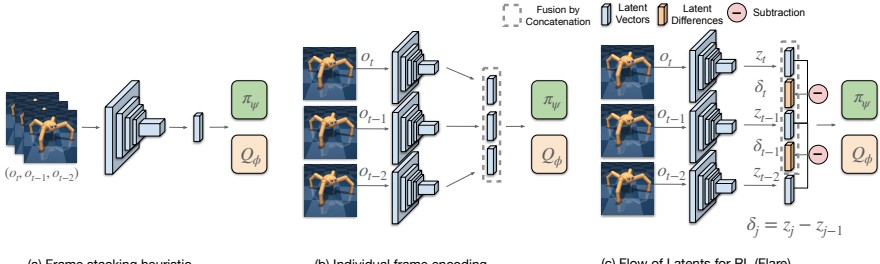

Figure 4: **F**low of **La**tents for **Re**inforcement Learning (*Flare*): (a) the architecture for the frame stacking heuristic, (b) an alternative to the frame stacking hueristic by encoding each image individually, and (c) the Flare architecture which encodes images individually, computes the feature differences, and fuses the differences together with the latents.

## 5.1 Latent Flow

In computer vision, the most common way to explicitly inject temporal information of a video sequence is to compute dense optical flow between consecutive frames [38]. Then the RGB and the optical flow inputs are individually fed into two streams of encoders and the features from both are fused in the later stage of the pipelinee. But two-stream architectures with optical flow are not as applicable to RL, because it is too computationally costly to generate optical flow on the fly.

To address this challenge and motivated by experiments in Section 4, we propose an alternative architecture that is similar in spirit to the two-stream networks for video classification. Rather than computing optical flow directly, we approximate temporal information in the latent space. Instead of encoding a stack of frames at once, we use a frame-wise CNN to encode each individual frame. Then we compute the differences between the latent encodings of consecutive frames, which we refer to as *latent flow*. Finally, the latent features and the latent flow are fused together through concatenation before getting passed to the downstream RL algorithm. We call the proposed architecture as **F**low of **La**tents for **Re**inforcement Learning (*Flare*).

## 5.2 Implementation and Architecture Details

---
**Algorithm 1** Pixel-based Flare Inference

  **Given** $\pi_\psi, f_{\text{CNN}}$
  **for** each environment step $t$ **do**
    $z_j = f_{\text{CNN}}(o_j), j = t-k, .., t$
    $\delta_j = z_j - z_{j-1}, j = t-k+1, .., t$
    $\mathbf{z}_t = (z_{t-k+1}, \cdots, z_t, \delta_{t-k+1}, \cdots, \delta_t)$
    $\mathbf{z_t} = \text{LayerNorm}(f_{\text{FC}}(\mathbf{z_t}))$
    $a_t \sim \pi_\psi(a_t|\mathbf{z}_t)$
    $o_{t+1} \sim p(o_{t+1}|a_t, \mathbf{o}_t = (o_t, o_{t-1}..o_{t-k}))$

---

We select RAD as the base algorithm to elaborate the execution of Flare. Similar adaptations can be seamlessly applied to other RL algorithms such as Rainbow DQN. The RAD architecture, shown in Figure 4a, stacks multiple data augmented frames observed in the pixel space and encodes them altogether through an CNN. This can be viewed as a form of early fusion [24].

Another preprocessing option is to encode each frame individually through a shared frame-wise encoder and perform late fusion of the resulting latent features, as shown in Figure 4b. However, we find that simply concatenating the latent features results in inferior performance when compared to the frame stacking heuristic, which we further elaborate in Section 6.3. We conjecture that pixel-level frame stacking benefits from leveraging both the CNN and the fully connected layers to process temporal information, whereas latent-level stacking does not propagate temporal information back through the CNN encoder.

Based on this conjecture, we explicitly compute the latent flow $\delta_t = z_t - z_{t-1}$ while detaching the $z_{t-1}$ gradients when computing $\delta_t$. We then fuse together $(\delta_t, z_t)$. Next, since negative values in the fused latent embedding now possesses semantic meaning from $\delta_t$, instead of ReLU non-linearity, we pass the embedding through a fully-connected layer followed by layer normalization, before entering the actor and critic networks as shown in Figure 4c. Pseudocode illustrates inference with Flare in Algorithm 5.2; during training, the encodings of latent features and flow are done in the same way except with augmented observations.

## 6 Experiments

In this section, we first present the main experimental results, where we show that Flare achieves substantial performance gains over the base algorithm RAD [26] and Rainbow DQN [19]. Then we conduct a series of ablation studies to stress test the design choices of the Flare architecture.

### 6.1 Environments and Evaluation Metrics

**The DeepMind Control Suite** (DMControl) [42], based on MuJoCo [43], is a commonly used benchmark for continuous control from pixels. On simpler environments in the suite, prior works [25, 26] have made substantial progress on this benchmark and closed the gap between state-based and pixel-based efficiency. However, on more challenging environments that feature partial observability, sparse rewards, or precise manipulation, these algorithms struggle to learn optimal policies efficiently . In this work, we focus on 5 of these more challenging tasks. The 5 environments include Walker

Table 1: Mean returns and standard errors on 5 challenging DMControl tasks evaluated at 500K and 1M environment steps over 5 random seeds and 10 trajectories per seed. Flare substantially outperforms RAD on a majority (3 out of the 5) of environments, while remaining competitive in the remaining ones and achieving a substantially higher aggregate scores.

| | 1M STEPS | | | 500K STEPS | | |
|---|---|---|---|---|---|---|
| TASK | FLARE | RAD | CURL | FLARE | RAD | CURL |
| QUADRUPED WALK | $\mathbf{488} \pm 99$ | $322 \pm 102$ | $38 \pm 10$ | $\mathbf{296} \pm 62$ | $206 \pm 50$ | $39 \pm 22$ |
| PENDULUM SWINGUP | $\mathbf{809} \pm 14$ | $520 \pm 144$ | $151 \pm 48$ | $\mathbf{242} \pm 68$ | $79 \pm 33$ | $46 \pm 207$ |
| HOPPER HOP | $\mathbf{217} \pm 26$ | $211 \pm 12$ | $44 \pm 3$ | $\mathbf{90} \pm 25$ | $40 \pm 18$ | $10 \pm 17$ |
| FINGER TURN HARD | $\mathbf{661} \pm 141$ | $249 \pm 44$ | $222 \pm 14$ | $\mathbf{282} \pm 30$ | $137 \pm 44$ | $207 \pm 32$ |
| WALKER RUN | $556 \pm 42$ | $\mathbf{628} \pm 17$ | $323 \pm 35$ | $426 \pm 18$ | $\mathbf{547} \pm 21$ | $245 \pm 32$ |
| AVERAGE RETURN | $\mathbf{546}$ | $388$ | $156$ | $\mathbf{267}$ | $202$ | $109$ |

Table 2: Due to large computational requirements for 100M Atari runs, we randomly select 8 Atari games for evaluation. We run 5 random seeds for both Flare and Rainbow DQN [19], evaluate scores at 100m training steps, and show the mean and standard error. Flare improves the Rainbow DQN in most games and achieves a subtantially higher mean and median human normalized scores (HNS). † refers to a comparison being made between Flare and Flare's base algorithm Rainbow. Reference values for DQN, Random, and Human baselines are taken from Bellemare et al. [4].

| TASK | FLARE | RAINBOW | DQN | RANDOM | HUMAN |
|---|---|---|---|---|---|
| DEFENDER | $\mathbf{86982} \pm 13065$ | $44694 \pm 1782$ | $23633$ | $2874.5$ | $18688.9$ |
| PHOENIX | $\mathbf{60974} \pm 8070$ | $16992 \pm 1474$ | $8485.2$ | $761.4$ | $7242.6$ |
| BERZERK | $\mathbf{2049} \pm 188$ | $1636 \pm 267$ | $585.6$ | $123.7$ | $2630.4$ |
| MONTEZUMA | $\mathbf{1668} \pm 472$ | $900 \pm 161$ | $0$ | $0$ | $4753.3$ |
| ASSAULT | $12724 \pm 221$ | $\mathbf{15229} \pm 1611$ | $4280.5$ | $222.4$ | $742$ |
| BREAKOUT | $\mathbf{345}^{\dagger} \pm 10$ | $280 \pm 8$ | $\mathbf{385.5}$ | $1.7$ | $30.5$ |
| SEAQUEST | $13901 \pm 3616$ | $\mathbf{24090} \pm 5579$ | $5860.6$ | $68.4$ | $42054.7$ |
| TUTANKHAM | $\mathbf{248} \pm 9$ | $\mathbf{247} \pm 5$ | $68.1$ | $11.4$ | $167.6$ |
| MEDIAN HNS | $\mathbf{3.4}$ | $2.0$ | $0.8$ | $0.0$ | $1.0$ |
| AVERAGE HNS | $\mathbf{6.7}$ | $5.8$ | $3.0$ | $0.0$ | $1.0$ |

Run (requires maintaining balance with speed), Quadruped Walk (partially observable agent morphology), Hopper Hop (locomotion with sparse rewards), Finger Turn-hard (precise manipulation), and Pendulum Swingup (torque control with sparse rewards). For evaluation, we benchmark performance at 500K and 1M *environment steps* and compare against RAD.

**The Atari 2600 Games** [3] is another highly popular RL benchmark. Recent efforts have led to a range of highly successful algorithms [11, 19, 23, 15, 2] to solve Atari games directly from pixel. A representative state-of-the-art is Rainbow DQN (see Section 3). We adopt the official Rainbow DQN implementation [31] as our baseline and simply incorporate Flare while retaining all the other default settings, including hyperparameters and preprocessing. Note that the baseline Rainbow DQN's model architecture is also modified to the most comparable setup to that of Flare as described in Section 5.2, including increasing the number of last layer convolutional channels (to match the number of parameters) and adding a fully-connected layer plus layer normalization before the Q networks. We evaluate on a diverse subset of Atari games at 100M *training steps*, namely Assault, Breakout, Freeway, Krull, Montezuma Revenge, Seaquest, Up n Down and Tutankham, to assess the effectiveness of Flare.

**Evaluation and Training Protocol** It is a common symptom in RL that evaluations appear noisy. To ensure the most fair presentation of results, we follow this protocol: the main results in Section 6.2 report the mean over 5 random seeds with standard error, a standard RL practice [42, 31]. Furthermore, we use the same 5 seeds for both the baseline and Flare.

## 6.2 Main Results

We present the main results of comparing Flare against the baselines, namely RAD and Rainbow DQN. Flare outperforms the baselines on the majority of the environments. It is worth noting that since these baselines already produce state-of-the-art level performances, any steady improvement under our rigorous experimental protocol—even when the gain seems minor —is significant.

**DMControl:** Our main experimental results on the 5 DMControl tasks are presented in Table 1. We find that Flare outperforms RAD in terms of both final performance and sample efficiency for majority (3 out of 5) of the environments, while being competitive on the remaining environments. Specifically, Flare attains similar asymptotic performance to state-based RL on Pendulum Swingup, Hopper Hop, and Finger Turn-hard. For Quadruped Walk, a particularly challenging environment due to its large action space and partial observability, Flare learns much more efficiently than RAD and achieves a higher final score. Moreover, Flare outperforms RAD in terms of sample efficiency on all of the core tasks except for Walker Run. The 500k and 1M environment step evaluations in Table 1 show that, on average, Flare achieves **1.4×** and **1.3×** higher scores than RAD at the 1M step and the 500K step benchmarks, respectively.

**Atari:** The results on the 8 Atari games are in Table 2. Again, we observe substantial performance gain from Flare on the majority (5 out of 8) of the games, including the challenging Montezuma's Revenge. On most of the remaining games, Flare is equally competitive except for Seaquest. In the appendix, we also show that Flare performs competitively when comparing against other DQN variants at 100M training steps, including the original Rainbow implementations.

## 6.3 Ablation Studies

We ablate a number of components of the Flare architecture on the Quadruped Walk and Pendulum Swingup environments to stress test the Flare architecture. The results shown in Figure 5 aim to answer the following questions:

**Q1**: *Do we need latent flow or is computing pixel differences sufficient?*
**A1**: Flare proposes a late fusion of latent differences with the latent embeddings, while a simpler approach is an early fusion of pixel differences with the pixel input, which we call pixel flow. We compare Flare to pixel flow in Figure 5 (left) and find that pixel flow is above RAD but significantly less efficient and less stable than Flare, particularly on Quadruped Walk. This ablation suggests that late fusion temporal information after encoding the image is preferred to early fusion.

**Q2**: *Are the gains coming from latent flow or individual frame-wise encoding?*
**A2**: We address the potential concern that the performance gain of Flare stems from the frame-wise ConvNet architectural modification instead of the fusion of latent flow. Concretely, we follow the exact architecture and training as Flare, but instead of concatenating the latent flow, we concatenate each frame's latent vector after the convolution encoders directly as described in Figure 4b. This ablation is similar in spirit to the state-based experiments in Figure 3. The learning curves in Figure 5 (center) show that individual frame-wise encoding is not the source of the performance lift: frame-wise encoding, though on par with RAD on Pendulum Swingup, performs significantly worse on Quadruped Walk. Flare's improvements over RAD are hence most likely thanks to the explicit fusion of latent flow.

**Q3**: *How does the input frame count affect performance?*
**A3**: We compare stacking 2, 3, and 5 frames in Flare in Figure 5 (right). We find that changing the number of stacked frames does not significantly impact the locomotion task, quadruped walk, but Pendulum Swingup tends to be more sensitive to this hyperparameter. Interestingly, the optimal number of frames for Pendulum Swingup is 2, and more frames can in fact degrade Flare's performance, indicating the immediate position and velocity information is the most critical to learn effective policies on this task. We hypothesize that Flare trains more slowly with increased frame count on Pendulum Swingup due to the presence of unnecessary information that the actor and critic networks need to learn to ignore.

**Q4**: *Do we need latent flow or is RNN over latent sufficient?*
**A4**: Another approach of latent fusion would be applying a recurrent neural network on the latent embeddings. We compare FLARE with LSTM baselines on DMControl. We found that RNNs

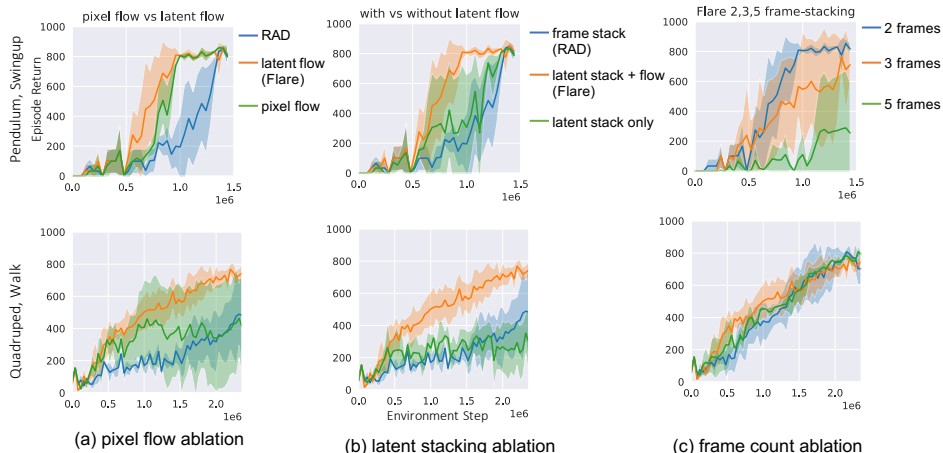

(a) pixel flow ablation     (b) latent stacking ablation     (c) frame count ablation

Figure 5: We perform 3 ablation studies: (a) *Pixel flow ablation*: we compare using pixel-level and latent-level (Flare) differences. Flare is more stable and performs better. (b) *Latent stack ablation*: we compare using latent stack with and without the latent flow. The latter performs significantly worse, suggesting that the latent flow is crucial. (c) *Frames count ablation*: we test using different number of frames for Flare.

perform worse than FLARE (see Fig 6), which is in agreement with our findings over the poor performance of RNN on coordinate (i.e. compact) state representations in Section 4.

## 7 Broader Impacts and Limitations

**Conclusion** We propose Flare, an architecture for RL that explicitly encode temporal information by computing flow in the latent space. In experiments, we show that in the state space, Flare can recover the optimal performance with only state positions and no access to the state velocities. In the pixel space, Flare improves upon the state-of-the-art model-free RL algorithms on the majority of selected tasks in the DMControl and Atari suites, while matching in the remaining. All code assets used for this project came with MIT licenses. For more details, we refer the reader to the appendix.

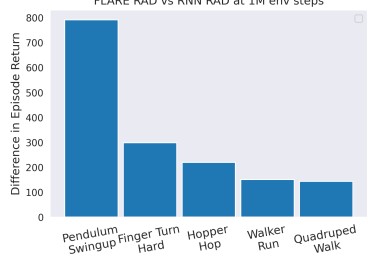

Figure 6: The bars show the difference in episode return between FLARE RAD and RNN at 1M env steps. All experiments are run on a fixed seed. FLARE outperform the RNN baseline in all of the experiments

**Limitations** Flare is a general approach to improve RL algorithms on many environments and tasks but not the panacea to single-handedly solve all. For instance, learning to control humanoid from pixels remains a challenge even when augmenting RAD with Flare. Also, RAD without Flare in fact is preferred for Seaquest, one of the Atari games. An additional limitation is that Flare is only useful if temporal information such as velocities is visible in the input pixel images and may not work as well for partially observed environments. Finally, Flare was only tested on model-free algorithms and it would be informative to investigate its applicability in the model-based regime, which we leave for future work.

**Broader Impacts** Simple architectures that improve performance can be impactful due to their ease of use and wide applicability. Prime examples of such innovations in vision are ResNets [18] which have been widely adopted. A potential negative consequence of Flare and supervised RL algorithms in general is that they rely on hand-designed reward functions which can be exploited. For example, in the Quadruped task Flare learns a more optimal policy than prior model-free methods but the resulting policy is extremely jittery since the simulator does not penalize jerk. This policy

would likely wear out the joints if deployed on real robot. To make architectures like Flare and RL in general more applicable to real-world scenarios it would be useful to investigate how to generate safe policies perhaps through constrained optimization or offline RL.

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
