# OpenReview forum: "Reinforcement Learning with Latent Flow"
_NeurIPS.cc/2021/Conference — NeurIPS 2021 Poster_

### Official Review · Reviewer_odFP · 2021-06-30

**Rating:** 6
**Confidence:** 4

**Summary:**

The paper proposes a simple modiﬁcation to existing deep RL architectures when the input is a sequence of frames. It makes use of a specific encoding of the frames and a specific way of combining CNN and fusion of the information from the different frames. This architecture does not require any additional auxiliary tasks or supervision signals as compared to traditional RL losses. The results of the paper show that the developed approaches are significantly better than baselines.

**Limitations And Societal Impact:**

The limitations and broader impact are discussed in a suitable way.

**Main Review:**

The proposed technique is built on top of existing techniques such as Rainbow DQN and RAD and it achieves a gain in performance in many cases. The benchmarks considered are pixel-based continuous control tasks within the DeepMind control benchmark suite and Atari games.

The paper also provides different ablations, which contribute to understanding the importance of the different components of the proposed method.

This paper provides interesting insights about a late fusion architecture and a specific encoding in the context of RL from pixels. Even though the technique is quite simple and the improvements are in most cases not large (but still significant), the experiments and the ablations are conducted in good way with a few interesting baselines. The paper is also quite well-written and provides clear illustrations of the approach and of its results on the benchmarks.

Additional positive points: a source code is provided, the error bars are reported (except on Figure 6?).

Additional comment:
- there are quite a few typos: "Notebly", "signifcant", "an SAC variant", "pipelinee"
- a few sentences at the beginning of section 3 that defines the setting formally with o_t, a_t, etc. could be useful.

**Time Spent Reviewing:**

3

---

> ### Author Response · Authors · 2021-08-10
> **Response to Reviewer odFP**
>
> Thanks so much for spending a lot of time on reviewing this work and providing constructive assessment!
> We are glad that you found our experiments and ablations to be well-designed and insightful. We are also happy that you found the performance improvement to be significant. Thanks for taking a careful review and pointing out those typos.
>
> **Q1: Error bars are not reported on Figure 6**
> Figure 6 is presenting the difference of the average return between Flare and the RNN baseline. Since it is computing the difference, we did not present standard deviation here since it is not clear which standard deviation to show (FLARE or RNN). We do report standard deviations for FLARE and the RNN baseline individually at 1M env steps.
>
> | | Quadruped Walk| Pendulum Swingup | Hopper Hop | Finger Turn Hard | Walker Run |
> |--|--|--|--|--|--|
> |FLARE| 99| 14| 26| 141| 42|
> |RNN variant| 86| 10| 0.2| 70| 26|
>
> This way of plotting the difference of average return without error bars for comparing different algorithms is also used in the Dreamer v2 [1] paper (Fig E.1).
>
>
> References:
> [1] Hafner, Danijar et al. “Mastering Atari with Discrete World Models.” ArXiv abs/2010.02193 (2021): n. pag.

---

> > ### Comment · Reviewer_odFP · 2021-08-11
> > **The rebuttal addresses my minor concerns**
> >
> > The rebuttal addresses my minor concerns. After having also read the other reviews and all answers from the authors, I keep my score of 6 (marginally above the acceptance threshold) and recommend acceptance: even though the contribution might be considered as "simple", I think that the paper does not have major flaws and that it provides a valuable contribution.

---

### Official Review · Reviewer_oLLK · 2021-07-09

**Rating:** 6
**Confidence:** 4

**Summary:**

In this paper, the authors modify two state-of-the-art reinforcement learning algorithms (RAD-SAC and Rainbow-DQN) by changing the way information from multiple timesteps is aggregated. Instead of stacking frames at a pixel level, the authors propose to encode each frame individually and then pass the encodings and their pairwise differences to the actor/critic networks. It is shown that when learning policies from pixels this way of fusing information substantially outperforms the baseline variants in 3/5 environments from the DeepMind control suite and 5/8 environments from Atari2600. Furthermore, experiments with low-level observations on 5 DeepMind Control Suite tasks support the particular choice of information fusion over using recurrent networks or stacking states directly without including state differences.


**Limitations And Societal Impact:**

The authors sufficiently cover the limitations of their work. Societal impacts are not explicitly addressed. In my opinion, this work does not exhibit extraordinary societal impacts  in addition to those induced by basic reasearch on reinforcement learning methods in general.


**Main Review:**

**Originality:**
The proposed way to fuse information from multiple frames / timesteps is simple but novel. It questions the "standard way" of stacking frames to incorporate information from multiple timesteps into the RL algorithm, which is interesting.


**Quality:**

* The authors perform reasonable experiments to evaluate the advantage of their proposed method for low-dimensional and high-dimensional observations

* The motivation for selecting the subset of environments reported in Table 1 is unclear to me. In other work such as CURL, Dreamer and PlaNet, different environments and timesteps were selected (namely Finger Spin, Cartpole Swingup, Reacher Easy, Cheetah Run, Walker Walk, Cup Catch) at 100k steps / 500k steps. Evaluating FLARE on these environments and timestep limits would greatly simplify comparison with the above methods.

* I am concerned with the fairness of comparing the "stack frames" and "stack latents" variants to FLARE. In FLARE the "fused" (concatenated) vector of latents and differences is passed through a fully connected network before entering the actor and critic networks (Algorithm 1). Is this fully connected network also active for the "stack frames" / "stack latent" variants? If not, don't the FLARE actor/critic networks effectively have one more layer compared to the baseline variants? What happens when just z_t is passed to actor/critic, without f_FC and LayerNorm?


**Clarity:**
* The paper is well-written and easy to follow and sufficiently covers preliminary methods and related work. Figure 4 is very helpful to understand the proposed method fusioning information from multiple timesteps.

* The inputs to the "Stack SAC" baseline as denoted in the caption of Figure 3 does not align with Figure 4c. In the Stack SAC baseline, the current state and 3 differences comprise the input. In Figure 4, it is 3 states and 2 differences. Is this intended?

* The set of environments used for Fig. 2, 3 differs from the set of environments on which numerical results are reported (Table 2). Is there a particular reason for this?

* Checklist item 3d) is not fulfilled, I did not find a report on the type and amount of computational resources used.

* Questions:
  * l. 208: Why exactly do you detach z_{t-1} gradients?
  * l. 209: Why should a ReLU non-linearity not be used as activation function when negative input values possess semantic meaning?


**Significance:**
If further experiments and clarifications (see 'Weaknesses') support the claim that the proposed modifications to state-of-the-art model free RL algorithms substantially and consistently improve their performance this would be of high interest to the RL community.


**Conclusion:**
Although the methodological contribution is rather small it would be of interest for the RL community in my opinion, under the condition that the proposed modification substantially and consistently improves performance over the base variants RAD and RAINBOW. However, in my eyes, some experiments and clarifications are missing to support this claim. First, reporting (numerical) results on (Finger Spin, Cartpole Swingup, Reacher Easy, Cheetah Run, Walker Walk, Cup Catch) would greatly simplify comparison to other state-of-the-art RL algorithms. Second, I am concerned about the fairness of the comparison between baseline variants ("stack frames" / "stack latents") and FLARE (see 'Weaknesses'). For this reason, I do not see the paper ready for publication in NeurIPS yet.


**Minor comments:**
* In Table 1 in the appendix, "FLARE" is erroneously marked as bold for the "Berzerk" environment although Rainbow from "DQN Zoo" performs better

**Time Spent Reviewing:**

5

---

> ### Author Response · Authors · 2021-08-10
> **Response to Reviewer oLLK**
>
> Thanks so much for spending a lot of time on reviewing this work and providing constructive assessment!
> We are glad that you found our approach to be simple but novel, and acknowledge its potential to be of interest for the RL community. We are also happy that you find the paper to be well-written.
>
> **Q1: Why are results on environments, such as Finger Spin, Cartpole Swingup, Reacher Easy, Cheetah Run, Walker Walk, Cup Catch, not reported? How were the environments chosen?**
> Thank you for this suggestion. We did not include the environments you mention because they are considered solved - pixel-based learning on these environments is as efficient as learning from coordinate state so there is little room left for improvement. For this reason, we choose harder environments where there is still a gap between pixel-based and state-based efficiency. These environment choices are explained in Section 6.1 of the paper. We show that in the environments where the pixel-state efficiency gap has not been closed, Flare improves over the baseline.
>
> Nevertheless, you have a good point that we should also report results on the heavily-benchmarked environments for completeness. We therefore have additional experiments for Flare and the baseline RAD for the following common environments on the 100k/500k benchmark. We show in the table below that Flare is competitive with RAD on the saturated environments while outperforming it on the harder ones (see main paper). We will revise the appendix to include these results.
>
> |                    |FLARE (100k) |      RAD (100k)| FLARE (500k)|      RAD (500k)|
> |----------------|---------------|--------------------|----------------------|------------------|
> |Finger Spin| $253\pm{5}$ | $237\pm{7}$ | $699\pm{14}$ | $656\pm{31}$|
> |Reacher Easy|  $376\pm{22}$ | $375 \pm{52}$    | $887\pm{34}$ |$893\pm{16}$|
> |Walker Walk| $483\pm{7}$ |$507\pm{28} $          | $943\pm{19}$ |$932\pm{22}$|
> |Ball-in-cup Catch| $354\pm{209}$ |$331\pm{147}$   |  $961\pm{9}$ |$970\pm{11}$|
> |Cheetah Run| $213\pm{17}$ |$225\pm{37} $         | $547\pm{21}$ |$541\pm{10}$|
>
>
> **Q2: For Table 1, why not compare at 100K/500K steps as this is the standard?**
> The 100k / 500k benchmark is useful for easier tasks that are solvable within 500k environmental steps but harder tasks require more samples (1M or more)  to converge at the optimal policy. Since we evaluate on the harder environments, we benchmark them at 500k/1M steps.
>
>
> **Q3: Does "stack frames" and "stack latents" miss a fully connected and LayerNorm layer?
> What happens when just $z_t$ is passed to actor/critic, without $f_{FC}$ and LayerNorm?**
>
> We apologize for the confusing communication on our end. All the model-free baselines use the exact same architecture (including the FC and LayerNorm layers), which we clarify below:
>
> The framework figure (Fig 4) seems to be giving the impression that the “stacked latent” or “stacked flow” goes directly to the actor and critic. Yet, this is actually not the case, as explained in Section 5.1 in the paper. First, the latent stacking happens after the convolutional layers’ output and before the fully connected and LayerNorm layers. That is, output of the convolutional layers is flattened and then be used for stacking, which then goes through the fully connected and LayerNorm layers. Second, the latent in all baseline and FLARE are bottlenecked to be of 64 in dimension before passing to actor and critic to ensure the fairness of comparison. Thus, we need the FC layer to adjust dimension from the convolutional layers’ output to 64. In summary, all algorithms use the exact same architecture, including the fully connected and LayerNorm layers, as they are needed for dimension adjustment, and we can’t pass $z_t$ to actor/critic without FC and LayerNorm also due to dimension differences.
>
> That being said, there is indeed a parameter number difference between RAD and Flare. However, this is unavoidable because of this difference in dimension. Even though Flare has somewhat more parameters than RAD, we demonstrate the improvement is not coming from that by comparing “stack latents” with Flare, as “stack latents” and Flare have the same number of parameters and differ only in whether using flow or not. Again, thanks for pointing this out. We are sorry that we left out some details when trying to summarize the process into a figure and we will be sure to revise this figure.
>
> **Q4: Why exactly do you detach $z_{t-1}$ gradients?**
> We found the performance is worsen when $z_{t-1}$'s gradients is not detached. For Quadruped Walk at 1M env steps, FLARE has episode return of $488\pm{99}$, while not detaching $z_{t-1}$'s gradients has episode return of $331\pm{202}$
>
> **Q5: Why should a ReLU non-linearity not be used as an activation function when negative input values possess semantic meaning?**
> Thanks for pointing this out. The reason we don’t include a ReLU activation between the CNN encoder and the actor / critic MLPs is for clarity of comparison. This is a design choice from the main baseline RAD and we kept it in order to ensure the encoding architecture is similar. We will be sure to revise this sentence in our paper.

---

> > ### Comment · Reviewer_oLLK · 2021-08-13
> > **Response to authors**
> >
> > I thank the authors for their detailed response and the effort to conduct additional experiments. They have sufficiently responded to the concerns I raised in my review, explaining that their evaluation is fair and providing results on a "standard benchmark" set to compare to other methods. Overall, I think it is a well-executed paper over a small (but intuitive and reasonable) methodological contribution and therefore still a valuable contribution to the community.
> >
> > One last thing which concerns me a bit is that, given that the proposed method performs worse on some environments (e.g., Walker Run, Assault, Seaquest), I am not sure if the community will agree on the method as "the new standard" to do or rather it will double the hyperparameter search space for future RL experimentation (latent flow vs stack latents).
> >
> > Given these two points, I increase my score to "weak accept".

---

### Official Review · Reviewer_NvL5 · 2021-07-14

**Rating:** 7
**Confidence:** 4

**Summary:**

This paper proposes adding temporal information in the form of latent flow to deep RL agents. The idea is simple: temporal differences between latent embeddings of consecutive frames are computed; the differences are then concatenated to the embeddings for downstream processing by an RL controller. This idea is inspired by idea of late fusion in video processing and stands in contrast to the current approach of frame-stacking, which is more akin to early fusion. The authors have validated their approach on a series of DMC tasks as well as Atari games.

**Limitations And Societal Impact:**

The authors have given thoughtful consideration to the limitations of their work, including instances where their method may not work. I would have also appreciated some discussion of needing to tune the number of frames as a hyperparameter. The authors have also discussed the broader impact of their work.

**Main Review:**

Strengths:
* the paper is well-written and the ideas clearly laid out
* the method is easily applicable to many different RL architectures
* empirical results show a clear improvement over baselines (including RNN baselines) on a selection of DMC and Atari environments
* informative ablation studies comparing frame-stacking, temporal pixel differences, and pure latent stacking

Weaknesses:
* number of frames to include adds an extra hyperparameter that needs to be tuned. The appropriate number will probably be task-dependent. The authors have demonstrated one case where extra frames do nothing and another case where it actually hurts (presumably because the agent must learn to ignore the superfluous information). However, one can conceivably imagine cases where having extra frames is necessary (e.g. if accelerations are important, then using two frames won't be enough).

Minor typos:
* line 60: “Notably” instead of “Notebly”
* line 212: “Algorithm 1” instead of “Algorithm 5.2”
* line 238: different set of games listed here compared to the table

Overall, while the idea is simple, I believe the authors have demonstrated the method does benefit RL algorithms. I am also satisfied that the authors have addressed concerns raised in their previous submission regarding a missing RNN baseline. I recommend accepting this paper.




**Time Spent Reviewing:**

3

---

> ### Author Response · Authors · 2021-08-10
> **Response to Reviewer NvL5**
>
> Thanks for spending a lot of time on reviewing this work and providing constructive assessment! We are glad that you found our paper to be well-written and that you appreciate the simplicity of our method. Also, thanks for pointing out the typos!
>
> **Q1: Does number of frames become an extra hyperparameter that needs to be tuned?**
> In our ablation studies, we demonstrated that increasing the number of frames could hurt performance or have no influence. Thanks for pointing out there might be cases where increasing the number of frames would help, such as when two frames is not enough. To investigate this possibility, we reran the ablation on two more environments: Cartpole Swingup and Reacher Easy. Results are shown below, evaluated at 500k env steps.
>
> || 2 frames| 3 frames| 5 frames|
> |---|---|---|--|
> |Cartpole Swingup| $537\pm{40}$ | $797\pm{24}$ | $805\pm{33}$|
> |Reacher Easy| $936\pm{38}$ | $918\pm{75}$ | $941\pm{79}$ |
>
>
> On the easier Cartpole Swingup environment, more frames does not seem to hurt performance. Although two frames could learn the optimal policy, having more than two frames speeds up the learning significantly. On the other hand, for Reacher Easy, performance does not depend on the number of frames like on Quadruped Walk (Fig 5). Since increasing frames could hurt, improve or have no effect over performance, this shows that the number of frames indeed could be a hyperparameter that would be best to be tuned for some task.

---

### Official Review · Reviewer_FyFS · 2021-07-20

**Rating:** 5
**Confidence:** 5

**Summary:**

The paper suggests using an optical flow to encode pixel observations in contrast to simple concatenation over the temporal dimension. This proposed method enables efficient recovery of dynamics information such as velocity and acceleration, this in turn, improves performance on several image-based tasks from DMControl and Atari.

**Ethical Concerns:**

No issues

**Limitations And Societal Impact:**

Yes

**Main Review:**

Originality: I think the paper is limited in its originality. While the particular application of an optimal flow like architecture to this type of setup is novel, it still a rather incremental derivation from prior work (both in CV and RL). This should not be mistaken with the simplicity aspect of Flare, which I like.


Quality: The paper is well executed. The experimentation is sound and extensive. I like the fact that the authors consider several different domain (DMControl and Atari) and several different algorithm, which makes it easier to justify their method.

Clarity:
The paper is very well written and easy to follow. I like the motivation example and figures 2 and 3. I also like how the ablation section is structured as a Q&A, which makes it easy to comprehend.

Significance:
I believe the significance of this paper is limited. While it proposes a better way to deal with encoding temporal information in image-based RL, it feels more like a hack, rather than a systematic solution. For example, it is not hard to imagine scenarios where it is not enough to extract linear relationships between consecutive frames. In these cases more general architectures such as RNNs or Transformers will be still required.

Overall:
I think it is a well written and executed paper that demonstrates some interesting findings. I also like the fact that the authors attempt to investigate alternative architectures in RL. However, the novelty and significance aspects are limited. I'm thus borderline on this paper and biasing towards rejection.

**Time Spent Reviewing:**

2

---

> ### Author Response · Authors · 2021-08-10
> **Response to Reviewer FyFS**
>
> Thanks so much for spending a lot of time on reviewing this work and providing constructive assessment!
> We are glad that you found the experiments to be well-executed and extensive. We are also glad that you found the paper to be well-written and easy-to-follow.
>
> **Q1: As the method is incremental, this work has limited originality and significance.**
> We are delighted that you appreciate the simplicity of our approach, but don’t think that it’s fair to conclude that a simple method is not novel or significant. For instance, ResNet [1] proposed a simple modification to DL architectures by adding skip connections and was impactful. Similarly, the baseline method used in this paper, RAD [2], proposed using data augmentations in RL and is now widely used as a baseline in pixel-based RL. In FLARE, we show improvement over existing RL baselines with a simple architectural modification that incorporates latent flow. To the best of our knowledge, we are the first to propose latent flow and show that on average it improves performance of the baselines and therefore think the contribution is novel.
>
> References:
> [1] He, Kaiming et al. “Deep Residual Learning for Image Recognition.” 2016 IEEE Conference on Computer Vision and Pattern Recognition (CVPR) (2016): 770-778.
> [2] Laskin, M. et al. “Reinforcement Learning with Augmented Data.” ArXiv abs/2004.14990 (2020): n. pag.

---

### Decision · Program_Chairs · 2021-09-27

**Decision:**

Accept (Poster)

**Comment:**

The paper proposes a simple but effective method to increase sample efficiency of RL in environments with pixel-based observations, by capturing temporal information via differences of consecutive image observation encodings. While, like framestacking, the method is expected to be needed and effective only in environments where capturing a non-trivial observation history is necessary for good performance and despite the performance advantage over alternatives being moderate, the experiments contain an informative ablation study that makes this work useful to other researchers from this field.